# POSS Engineering of Multifunctional Nanoplatforms for Chemo-Mild Photothermal Synergistic Therapy

**DOI:** 10.3390/ijms25021012

**Published:** 2024-01-13

**Authors:** Zhengye Gu, Xiaochuan Geng, Shanyi Guang, Hongyao Xu

**Affiliations:** 1State Key Laboratory for Modification of Chemical Fibers and Polymer Materials, College of Materials Science, Engineering & Research Center for Analysis and Measurement, Donghua University, Shanghai 201620, China; guzhengye@126.com; 2Department of Radiology, Renji Hospital, Shanghai Jiao Tong University School of Medicine, Shanghai 200127, China; gengxiaochuan1949@126.com; 3College of Chemistry and Chemical Engineering, Donghua University, Shanghai 201620, China

**Keywords:** chemo-mild photothermal, synergistic effect, POSS-based nanoplatform, squaraine, high inhibition rate

## Abstract

Chemo-mild photothermal synergistic therapy can effectively inhibit tumor growth under mild hyperthermia, minimizing damage to nearby healthy tissues and skin while ensuring therapeutic efficacy. In this paper, we develop a multifunctional study based on polyhedral oligomeric sesquisiloxane (POSS) that exhibits a synergistic therapeutic effect through mild photothermal and chemotherapy treatments (POSS-SQ-DOX). The nanoplatform utilizes SQ-N as a photothermal agent (PTA) for mild photothermal, while doxorubicin (DOX) serves as the chemotherapeutic drug for chemotherapy. By incorporating POSS into the nanoplatform, we successfully prevent the aggregation of SQ-N in aqueous solutions, thus maintaining its excellent photothermal properties both in vitro and in vivo. Furthermore, the introduction of polyethylene glycol (PEG) significantly enhances cell permeability, which contributes to the remarkable therapeutic effect of POSS-SQ-DOX NPs. Our studies on the photothermal properties of POSS-SQ-DOX NPs demonstrate their high photothermal conversion efficiency (62.3%) and stability, confirming their suitability for use in mild photothermal therapy. A combination index value (CI = 0.72) verified the presence of a synergistic effect between these two treatments, indicating that POSS-SQ-DOX NPs exhibited significantly higher cell mortality (74.7%) and tumor inhibition rate (72.7%) compared to single chemotherapy and mild photothermal therapy. This observation highlights the synergistic therapeutic potential of POSS-SQ-DOX NPs. Furthermore, in vitro and in vivo toxicity tests suggest that the absence of cytotoxicity and excellent biocompatibility of POSS-SQ-DOX NPs provide a guarantee for clinical applications. Therefore, utilizing near-infrared light-triggering POSS-SQ-DOX NPs can serve as chemo-mild photothermal PTA, while functionalized POSS-SQ-DOX NPs hold great promise as a novel nanoplatform that may drive significant advancements in the field of chemo-mild photothermal therapy.

## 1. Introduction

As a highly promising candidate, photothermal therapy (PTT) has garnered significant attention in recent years. PTT utilizes light and photothermal agents (PTAs).

To eradicate cancer cells through photo-thermal conversion, offering higher efficiency, minimal invasiveness, and fewer side effects [1,2,3,4,5,6,7,8,9,10,11,12]. However, it typically relies on local hyperthermia (above 50 °C) [13,14,15,16] for complete tumor ablation, which can result in damage to the surrounding healthy tissues, inflammation, and other undesirable biological effects during cancer treatments [17,18,19,20,21]. Therefore, achieving cancer cell death at mild temperatures is crucial for clinical practice [21,22]; thus, the concept of mild photothermal therapy (MPTT) has emerged. Nevertheless, in most cases, MPTT alone exhibits an inferior anticancer ability due to the presence of heat shock proteins (HSPs), which may repair cell apoptosis induced by a lower heating temperature such as 45 °C [23]. Hence, adjuvant strategies, including PTAs for the inhibition of HSPs [24], autophagy modulation [25,26,27], organelle targeting [28,29,30], and gas sensitization [31,32,33] are necessary to enhance the therapeutic effect of MPTT. In addition to these aforementioned strategies, with the assistance of nanotechnology, PTAs in MPTT and other agents like chemodrugs, radiosensitizers, and photosensitizers can form a nanoplatform with multimodal anti-tumor therapeutic effects [34,35,36,37]. On one hand, the tumor cell-killing performance of other treatments in combination with MPTT is improved. On the other hand, other treatments can sensitize tumor cells to be more vulnerable to mild heat, thus enhancing the tumor eradication effect of MPTT [21].

As one of the main strategies for cancer treatment, chemotherapy plays an important role in clinical cancer treatment. However, the non-specific distribution and resistance of chemotherapeutic drugs significantly limit the clinical applications of chemotherapy. The combination of MPTT with chemotherapy can solve these problems easily. First, MPTT may enable spatiotemporally controlled drug delivery by adjusting the position, power density, and working time of NIR light. Second, its mild temperature can make cancer cells more accessible and vulnerable to chemotherapeutic drugs. Third, the heat generated by PTAs can stimulate intratumoral penetration and the accumulation of drugs due to the increased tumor blood flow and vascular permeability [21]. In turn, enhanced chemotherapy efficacy can compensate for insufficient heat damage due to the mild temperature. Therefore, a more excellent anti-tumor effect can be achieved by combining MPTT with chemotherapy.

Squaraine dyes (SQs) are a well-studied class of zwitterionic dyes with stable quinoid structures and high planarity [38]. Typically, they consist of an electron-accepting squaric acid and two electron donors in a D-A-D pattern. Due to their advantageous structure, these dyes exhibit exceptional optical properties, including elevated photoconductivity, well-defined absorption in the visible and near-infrared regions, a high molar absorption coefficient, and impressive photothermal conversion efficiency (PCE) [39,40]. As a result, SQs find wide applications in various biological fields, such as biosensing [41], bioimaging [42,43], and phototherapy [44,45]. However, on account of the strong dipole–dipole and π–π stacking interactions between molecules, SQs tend to aggregate in aqueous solutions, which can negatively impact their photothermal performance and photostability [46]. Therefore, there is a challenge when using SQs for biological applications, especially in PTT, as follows: preventing the easy formation of dye aggregates while preserving their photophysical characteristics. 

The emergence of polyhedral oligomeric sesquisiloxane (POSS) can effectively address this issue. A typical POSS molecule possesses a three-dimensional cage-like structure consisting of a cubic silica core and eight surrounding organic corner groups. These corner groups can be readily modified into various functional groups, facilitating the convenient preparation of multifunctional polymers with POSS cores [47,48,49]. Moreover, due to the nanoscale size of POSS, diverse types of functional POSS-based nanoplatforms can be constructed. Additionally, Wang et al. reported that conjugating porphyrin to POSS resulted in increased photodynamic efficiency and reduced aggregation [50]. Consequently, incorporating POSS can mitigate the aggregation caused by π–π stacking interactions between molecules. Furthermore, based on our previous research conducted in our laboratory [51,52], we discovered that introducing PEG could effectively enhance water solubility and achieve amphipathy for POSS-based nanoparticles, thereby significantly improving cell permeability and the biocompatibility of these nanoplatforms while simultaneously enhancing their potential applications in biological imaging [53], drug delivery [54,55], and tissue regeneration [56].

Hence, based on the aforementioned introduction, a multifunctional POSS-based nanoplatform (POSS-SQ-DOX) was developed for mild chemo-photothermal synergistic tumor therapy. Firstly, POSS-PEG was synthesized via click chemistry to enhance the cell permeability and biocompatibility of the nanoplatform. Subsequently, SQ-N and DOX were incorporated into POSS using the same method. In this nanoplatform, SQ-N exhibited excellent photothermal properties and high PCE and served as a PTA in MPTT. Additionally, DOX acted as a chemotherapeutic drug for chemotherapy purposes. As illustrated in Figure 1, upon in situ injection, POSS-SQ-DOX NPs accumulated within tumor cells where the mild thermal effect induced by near-infrared light at 808 nm worked synergistically with DOX to achieve an enhanced anti-tumor effect. The experimental results demonstrated that tumor growth was significantly inhibited in mice treated with POSS-SQ-DOX NPs compared to those receiving single DOX chemotherapy or MPTT alone. Furthermore, this treatment approach proved safe while ensuring highly effective anti-tumor outcomes. Overall, this study explored the synergistic anti-tumor effect of DOX chemotherapy and photothermal therapy working under mild temperatures, highlighting the great potential for utilizing the chemo-mild photothermal synergistic therapy system based on DOX and SQ-modified POSS-based nanoparticles for tumor treatments, bringing a novel strategy to combat tumors.

## 2. Results and Discussion

### 2.1. Preparation and Characterization of POSS-SQ-DOX NPs

PEG, SQ-N, and DOX were all chemically attached to POSS nanoparticles through click chemistry, resulting in the formation of POSS-SQ-DOX NPs (Appendix A). The structures of these compounds were confirmed using ^1^H NMR and FITR (Appendix A). The structure of POSS-SQ-DOX NPs is shown in Figure 1a. Additionally, transmission electron microscope (TEM) and dynamic laser light scattering (DLS) techniques were employed to verify the nanostructures of POSS-SQ-DOX NPs. TEM analysis revealed that the size of POSS-SQ-DOX NPs was approximately 25 nm, while DLS measurements showed a size of 24 nm, which are both larger than that of pure POSS-SH [57] (Figure 1b,c). Furthermore, the particle size stability of POSS-SQ-DOX NPs was monitored over a two-week period (Figure 1d), demonstrating minimal changes in nanoparticle size and indicating excellent stability for at least two weeks. Additionally, UV-Vis absorption spectra and fluorescence spectra were used to investigate the optical properties of POSS-SQ-DOX NPs. As depicted in Figure 1e, distinct UV absorption peaks near 818 nm and a fluorescence emission peak at 949 nm were observed for POSS-SQ-DOX NPs. The presence of near-infrared absorption characteristics provided a foundation for utilizing POSS-SQ-DOX NPs as PTAs in mild photothermal therapy. Notably, compared to SQ-N alone, its incorporation into POSS nanoparticles significantly enhanced both absorption and emission intensities for SQ-N molecules within the hybrid system due to the increased intermolecular distances between SQ-N units and weakened π–π stacking interactions (Appendix A). Furthermore, no significant changes in either the position or intensity of the absorption peak were observed during storage at room temperature for two weeks, indicating that POSS-SQ-DOX NPs displayed optical stability under suitable conditions (Figure 1f). Moreover, contact angle measurements confirmed the enhanced hydrophilicity of POSS-SQ-DOX NPs. In comparison to POSS-SH, DOX, and SQ-N, POSS-SQ-DOX NPs exhibited superior hydrophilicity due to the PEG modification (Appendix A). In summary, we successfully synthesized NIR-absorbing and highly stable POSS-SQ-DOX NPs, making them a suitable and robust system for the following research.

### 2.2. Photothermal Performance of POSS-SQ-DOX NPs

In order to evaluate the photothermal performance of POSS-SQ-DOX NPs, an 808 nm laser was employed for irradiation, and the resulting temperature changes were recorded. As shown in Figure 2a, it was observed that the temperature increments of POSS-SQ-DOX NPs exhibited a positive correlation with their concentrations under the irradiation of an 808 nm laser (0.5 W/cm^2^). Specifically, when the concentration of POSS-SQ-DOX NPs reached 200 μg/mL, the temperature increased from 29.3 °C to 81.1 °C within 210 s (∆T = 51.8 °C). In addition, at a concentration of 50 μg/mL, the temperature elevation demonstrated an upward trend as the laser power densities increased (Figure 2b). These results collectively indicate that POSS-SQ-DOX NPs possess efficient light-to-thermal energy conversion capabilities and enable precise control over heat generation. Additionally, a high PCE value (η) of up to 62.3% was calculated for POSS-SQ-DOX NPs [58,59] (Figure 2c,d). Moreover, thermal images provided a more intuitive visualization of the temperature variations associated with POSS-SQ-DOX NPs (Figure 2e). In addition, significant chemosensitization effects occur when the tumor site temperature reaches 39.0–43.0 °C because, when the temperature is increased from 37.0 to 43.0 °C, the permeability of the cell membrane is significantly enhanced, which promotes the uptake of nano-drugs by cells [5]. Thus, a laser power of 0.5 W/cm^2^ and a concentration of 50 μg/mL were selected for the following experiments. Furthermore, photothermal conversion stability referred to their ability to withstand multiple excitation cycles without irreversible loss, which was an important parameter for characterizing organic dye molecules [60]. The stability of photothermal conversion was evaluated through heating–cooling cycle experiments. As shown in Figure 2f, the POSS-SQ-DOX NPs solution was irradiated with an 808 nm laser for 210 s and subsequently allowed to cool naturally to room temperature while recording the temperatures. It was evident that the temperature of POSS-SQ-DOX NPs exhibited negligible changes during five heating–cooling cycles, suggesting excellent stability and reproducibility of POSS-SQ-DOX NPs. Hence, all these experiments indicate that POSS-SQ-DOX NPs could serve as a highly efficient PTA in PTT due to their high PCE and good photochemical stability.

### 2.3. Evaluation Synergies Effect In Vitro

The cytotoxicity of different formulations (free DOX, free SQ-N, and different ratios of DOX and SQ-N) in HeLa cells was evaluated using Cell Counting Kit-8 (CCK-8) assays. Both single DOX and SQ-N +L exhibited cytotoxic effects on HeLa cells. However, the efficacy was suboptimal with an IC_50_ value of 19.94 μg/mL for DOX alone and 17.45 μg/mL for SQ-N +L (Figure 3a, Table 1). To determine the optimal ratio of DOX and SQ-N in nanoparticles, we assessed the impact of varying DOX contents on cell viability at a constant total concentration (Figure 3b). Furthermore, combination index (CI) values were calculated to confirm the synergistic effect between different ratios and CI < 1 indicated synergism. Our results demonstrate clear synergistic effects within the range of DOX:SQ-N +L = 1:3–3:1, with the strongest synergistic effect observed at a dose ratio of DOX:SQ-N +L was 2:1 (*n*/*n*), resulting in an IC_50_ value as low as 9.32 μg/mL for DOX and 4.33 μg/mL for SQ-N when prepared as POSS-SQ-DOX NPs. In summary, by utilizing a ration of DOX:SQ-N +L = 2:1 during preparation, not only could we ensure a synergistic therapeutic effect but also reduce chemotherapy drugs dosage while alleviating potential side effects.

### 2.4. In Vitro Tumor Therapy

It is essential to assess the biocompatibility of POSS-SQ-DOX NPs prior to conducting further in vitro and in vivo anti-tumor therapy experiments. Moreover, all experiments were performed under 808 nm laser irradiation with a power density of 0.5 W/cm^2^. The toxicity of POSS-SQ-DOX NPs was evaluated using the CCK-8 method. Figure 4a shows that even at a concentration of 200 μg/mL, the cell viability remained above 90% for POSS-SQ NPs, indicating their safety for both normal L929 cells and tumor HeLa cells. However, when incubated with POSS-SQ-DOX NPs or a free DOX treatment for 24 h, the viability of HeLa cells was only slightly above 40%, suggesting that the toxicity of POSS-SQ-DOX NPs primarily stemmed from the presence of the DOX drug (Figure 4b). Furthermore, as depicted in Figure 4c,d, it could be visually confirmed that chemo-mild photothermal synergistic therapy exhibited better efficacy compared to single chemotherapy or PTT alone. Additionally, the IC_50_ value for DOX was determined to be 10.94 μg/mL for POSS-SQ-DOX NPs, which is lower than the 19.94 μg/mL of free DOX. These results collectively indicate that chemo-mild photothermal synergistic therapy exerts a stronger inhibitory effect on HeLa cell proliferation with reduced amounts of chemotherapy drugs compared to single chemotherapy or PTT. This further supports the notion that chemo-mild photothermal synergistic therapy could achieve enhanced anti-tumor therapeutic effects. 

Additionally, live-dead cell-staining experiments were carried out to further confirm the synergistic anti-tumor effect of chemo-mild photothermal synergistic therapy (Figure 4e). In these experiments, living cells were stained with Calcein AM (green), and dead cells were stained with PI (red). Compared to the DOX group, the POSS-SQ-DOX NPs group exhibited enhanced cytotoxicity against HeLa cells due to the superior cell permeability of the nanoparticles. Furthermore, the PI-positive rate was calculated for each group. As shown in Figure 4f, the PI-positive rate of the POSS-SQ-DOX +L group was 1.9-fold higher than that of the POSS-SQ +L group and 1.7-fold higher than that of the POSS-SQ-DOX group, confirming that POSS-SQ-DOX NPs achieved effective cellular ablation under 808 nm laser irradiation. These findings collectively demonstrate how POSS-SQ-DOX NPs could effectively exert a synergistic effect in chemo-mild photothermal therapy by more efficiently inhibiting tumor cell proliferation and inducing apoptosis. 

### 2.5. Anti-Tumor Therapy In Vivo

To assess the feasibility of chemo-mild photothermal therapy in tumors, we established a HeLa tumor-bearing mouse model and utilized the following five formulations for our studies: PBS, DOX, POSS-SQ +L, POSS-SQ-DOX, and POSS-SQ-DOX +L. In subsequent experiments, the concentrations of DOX, POSS-SQ NPs, and POSS-SQ-DOX NPs were 12 µg/mL, 18 µg/mL, and 50 µg/mL, respectively. All experiments were carried out under irradiation from an 808 nm laser (0.5 W/cm^2^). Notably, significant thermal signals were observed in tumor areas following the injection of POSS-SQ NPs and POSS-SQ-DOX NPs, while there was no clear single detection with PBS or DOX alone (Figure 5a). Real-time temperature changes at the tumor site were also recorded to further validate these thermal imaging results (Figure 5b). Tumor volume change data were documented, as shown in Figure 5c–g. While single chemotherapy (DOX group) and mild photothermal therapy (POSS-SQ +L group) exhibited certain inhibitory effects on tumor growth, they still resulted in an increased tumor volume with only moderate therapeutic efficacy. However, the inhibitory effect on tumors was slightly better in the POSS-SQ-DOX group compared to the DOX group due to the enhanced cell permeability provided by the use of POSS-SQ-DOX NPs. Moreover, the smallest tumor volume was observed in the group of POSS-SQ-DOX +L, indicating that this combination had a more pronounced efficacy in suppressing tumor growth. Furthermore, in sharp contrast with other experimental groups, both the tumor weight and the tumor inhibition ratio (TIR) were meticulously recorded and calculated (Figure 5h,i). The TIR value of the POSS-SQ-DOX +L group was found to be 72.7%, which exhibited a remarkable increase of 2.35 times compared to DOX alone, which is 1.46 times higher than POSS-SQ +L and 1.53 times higher than POSS-SQ-DOX. Additionally, the changes in body weight of tumor-bearing mice in each group were recorded, and it was found that the weight change trend of different groups of tumor-bearing mice was similar. Furthermore, on day 12, the body weight of the tumor-bearing mice was similar between the experimental and control groups, which demonstrated the excellent biosafety profile of POSS-SQ-DOX NPs (Figure 5j). Based on these comprehensive findings, it can be inferred that chemo-mild photothermal therapy had superior tumor-suppressive effects when compared to single chemotherapy or photothermal therapy alone without any associated toxicity.

The therapeutic effects of all groups were further evaluated through H&E, Ki-67, and TUNEL staining (Figure 6a–c). According to the results of H&E staining, tumor cells in the PBS were in good condition. However, the DOX, POSS-SQ +L, and POSS-SQ-DOX groups showed varying degrees of tumor cell lysis, indicating that a single drug or mild photothermal therapy had inhibitory effects on the tumor. In contrast, extensive damage and destruction with significant cell lysis were observed in the POSS-SQ-DOX +L group. Moreover, compared to other groups, treatment with POSS-SQ-DOX NPs with laser irradiation resulted in a significant reduction in Ki67-positive cells and an increase in TUNEL-positive cells within tumors. This may be because the cell membrane integrity was severely disrupted under the synergistic treatment of chemotherapy and MPTT, which led to cell membrane lysis and nuclear erosion and ultimately resulted in the apoptosis of tumor cells (Appendix A). These findings confirm that chemo-mild photothermal synergistic therapy is superior to chemotherapy or photothermal alone. Consistently with previous studies, these results highlight the excellent anti-tumor effects of POSS-SQ-DOX NPs by inhibiting cell proliferation and promoting cell apoptosis. Overall, these results demonstrate that this system has an excellent synergistic effect and achieves an excellent anti-tumor therapeutic outcome. 

### 2.6. In Vivo Safety Evaluation

To further evaluate the in vivo biosafety of POSS-SQ-DOX NPs, histological analyses and hemanalysis were conducted. HeLa tumor-bearing mice were injected with POSS-SQ-DOX NPs at the tumor site and subsequently exposed to 808 nm laser irradiation at a power density of 0.5 W/cm^−2^ for 5 min. After 12 days post-irradiation, the major organs, including the hearts, livers, spleens, lungs, and kidneys, were dissected from the mice and stained with H&E (Figure 7a). The results revealed no discernible organ damage in the POSS-SQ-DOX +L-treated mice compared to those treated with PBS alone, indicating its excellent biocompatibility and minor side effects for anti-tumor therapy. To further evaluate the potential toxicity of POSS-SQ-DOX NPs in vivo, two groups of mice were sacrificed for a blood routine assay after 12 days of treatment. As shown in Figure 7b–g, there was no significant difference in the parameters of blood routines between the PBS group and the POSS-SQ-DOX +L group, and all data were within the normal range. Normal white blood cell counts and lymphocyte counts indicated that there was no infection or inflammation, and the normal red blood cell counts and platelet counts indicated that there was no severe tissue damage. Altogether, the POSS-SQ-DOX NPs displayed excellent biocompatibility and biosafety and could be applied in chemo-mild PTT against cancers.

## 3. Materials and Methods

### 3.1. General Methods

#### 3.1.1. Materials

1,8-naphthalenediamine, acetone, bismuth trichloride (BiCl_3_), squaraine (SQ), 2,2-dimethoxy-2-phenylacetophenone (DMAP), potassium carbonate, n-butanol, doxorubicin, toluene, triethylamine, ethanol (C_2_H_5_OH), acryloyl chloride, methylene chloride (CH_2_Cl_2_), triethylamine, dimethyl sulfoxide (DMSO) and tetrahydrofuran (THF) were supplied by Sinopharm Chemical Reagent Co., Ltd. (Beijing, China). without further purification. Calcein AM and propidium iodide were purchased from Beyotime Biotechnology (Shanghai, China). L929 cell lines and cervical cancer cell lines (HeLa) were obtained from the Cell Bank of the Chinese Academy of Sciences. Male BALB/c mice and nude mice at the age of 6 weeks were purchased from Shanghai Silaike Experimental Animal Co., Ltd. (Shanghai, China).

#### 3.1.2. Synthesis of DOX-AC

Doxorubicin (1.09 g, 2.0 mmol) was dissolved in 25 mL of dry dichloromethane in a three-necked round bottom flask with stirring. After dissolution, 3.0 mL of triethylamine was added to the solution, and the solution was placed in an ice–water bath with stirring. A solution of 0.35 mL (3.6 mmol) of acryloyl chloride in 12 mL of dichloromethane was placed dropwise into the flask for over 30 min. Then, the reaction lasted for 24 h at 25 °C and was monitored via the point plate. After the completion of the reaction, the solution was dried and concentrated using a rotary evaporator. The crude product was crystallized using column chromatography (silica gel, ethanol/chloroform = 1: 20), and after vacuum-drying for 8 h, the DOX-AC yielded a white solid. Yield: 69%. 1 H NMR (600 MHz, DMSO, 298 K, *δ*/ppm): *δ*10.2 (s, 1 H), 10.0 (s, 1 H), 9.3 (s, 1 H), 7.9 (m, 2 H), 7.7 (m, 2 H), 7.3 (d, *J* = 1.1 Hz 2 H), 7.2 (m, 3 H), 6.9 (m, 2 H), 6.8 (m, 2 H), 6.7 (d, *J* = 1.2 Hz, 2 H), 6.6 (m, 3 H), 6.5 (m, 3 H), 6.4 (dd, *J* = 8.6, 5.2 Hz, 2 H), 6.2 (dd, *J* = 5.2, 0.6 Hz, 2 H), 2.2 (s, 3 H). FTIR (KBr), v/cm^−1^: 3363 cm^−1^ (-OH); 2958, 2881 cm^−1^ (-CH_2_-), 1728 cm^−1^ (C=O), 1618 cm^−1^ (C=C), 1452 cm^−1^ (NH), 1121 cm^−1^ (C-O-C).

#### 3.1.3. Synthesis of POSS-SQ NPs

In total, 828.1 mg (0.50 mmol) of POSS-PEG and 14.0 mg (0.036 mmol) of DMPA were fully dissolved in 20 mL of THF in a three-necked flask filled with nitrogen (N_2_). Subsequently, 10 mL of anhydrous THF containing 288.0 mg (0.50 mmol) of SQ-AC was added to the flask. Then, the mixture was irradiated with UV light for 2 h. After the reaction, the solvent was removed via rotary evaporation, and the product was obtained as a viscous liquid. Yield: 100%. 1 H NMR (600 MHz, CDCl_3_, 298 K, *δ*/ppm): *δ* 11.1 (s, 2 H), 9.2 (s, 2 H), 8.0–8.0 (ddd, *J* = 0.3, 3.6 Hz, 2 H), 7.56 (m, 3 H), 7.21 (m, 2 H), 7.1 (m, 2 H), 6.7 (d, *J* = 1.1 Hz, 2 H), 6.6 (m, 4 H), 6.5 (m, 6 H), 6.3 (m, 5 H), 6.0 (dd, *J* = 0.6, 5.2 Hz, 2 H), 3.4 (q, *J* = 3.5 Hz, 12 H), 1.2 (t, *J* = 3.5 Hz, 18 H). FTIR (KBr), v/cm^−1^: 3411 cm^−1^ (-NH), 2923, 2846 cm^−1^ (-CH_2_-), 2569 cm^−1^ (-SH), 2186 cm^−1^ (C≡N), 1742 cm^−1^ (C=O), 1613 cm^−1^ (C=C), 1440 cm^−1^ (Ar-NH), 1149 cm^−1^ (Si-O).

#### 3.1.4. Synthesis of POSS-SQ-DOX NPs

For the synthesis of POSS-SQ-DOX NPs, up to 1.11 g (0.5 mmol) of POSS-SQ and 0.028 g (0.072 mmol) of DMPA were fully dissolved in 20 mL of THF in a three-necked flask filled with nitrogen (N_2_). Subsequently, 10 mL of anhydrous THF containing 0.598 g (1.0 mmol) of DOX-AC was added into the flask. Then, the mixture was irradiated with UV light for 2 h. After the reaction, the solvent was removed via rotary evaporation, and the product was obtained as a viscous liquid. Yield: 100%.1H NMR (600 MHz, CDCl_3_, 298 K, *δ*/ppm): *δ* 10.2 (s, 2 H), 10.1 (s, 3 H), 9.3 (s, 2 H), 8.2 (s, 6 H), 8.0 (m, 10 H), 7.7 (qt, *J* = 3.5, 6.7 Hz, 8 H), 7.5 (m, 10 H), 7.4 (td, *J* = 2.4, 3.7 Hz, 10 H), 7.2 (m, 10 H), 6.9 (m, 10 H), 6.7 (d, *J* = 1.2 Hz, 3 H), 6.6 (m, 12 H), 6.5 (m, 12 H), 4.3 (m, 18 H). FTIR (KBr), v/cm^−1^: 3453 cm^−1^ (-OH), 2971, 2869 cm^−1^ (-CH_2_-), 2538 cm^−1^ (-SH), 2367 cm^−1^ (C≡N), 1715 cm^−1^ (C=O), 1630 cm^−1^ (C=C), 1488 cm^−1^ (Ar-NH), 1216 cm^−1^ (C-O-C), 1132 cm^−1^ (Si-O). ESI-MS: *m*/*z*: 4670.57 [M+H]+, found 4671.57.

### 3.2. Characterization of POSS-SQ-DOX NPs

^1^H NMR spectra were recorded with a Bruker AVANCE III-HD 600 MHz NMR spectrometer (Bruker, Billericca, MA, USA). Fourier transform infrared (FTIR) spectra were obtained using a Thermo Nicolet 8700 spectrometer (Thermo Scientific, Waltham, MA, USA). The morphologies of POSS-SQ-DOX NPs were observed using transmission electron microscopy (TEM, Thermo Scientific, Waltham, MA, USA) with an accelerating voltage of 120 kV. The size distribution of POSS-SQ-DOX NPs was assessed via dynamic light scattering (DLS, Brookhaven Instruments Co., New York, NY, USA). Ultraviolet–visible (UV–vis) absorption spectra were measured with a Lambda 950 UV–vis spectrometer (PerkinElmer, Waltham, MA, USA).

### 3.3. Stability of POSS-SQ-DOX NPs

To evaluate the nanoparticles’ stability, POSS-SQ-DOX NPs were placed at 37 °C in PBS, and the sizes were detected at different times using DLS.

### 3.4. Optical Performance of POSS-SQ-DOX NPs

The optical performance of POSS-SQ-DOX NPs was evaluated using UV-Vis and fluorescence spectra. The UV scanning spectra were in the range of 600–1000 nm of the POSS-SQ-DOX NP solution (200 μg/mL), and the excitation wavelength was 850 nm of fluorescence spectra. Furthermore, the optical stability of POSS-SQ-DOX NPs was verified by measuring the solution after the dilution process, with storage for 7 days and storage for 14 days at 37 °C.

### 3.5. Cytotoxicity and Combination Index Evaluation

CCK-8 assays were carried out to evaluate the in vitro cytotoxicity and killing effect of SQ-N, DOX, POSS-SQ NPs, and POSS-SQ-DOX NPs on the L929 cell line and HeLa cell line (accession number: SCSP-504). After seeding cells in a 96-well plate (1 × 10^4^ cells/well) for 18 h in a humidified 5% CO_2_ atmosphere at 37 °C, the culture medium was replaced by a serum-free medium containing a series different concentrations of SQ-N, DOX, POSS-SQ NPs and POSS-SQ-DOX NPs. After incubation for another 8 h, half of the parallel wells were exposed to 808 nm laser irradiation (0.5 W/cm^2^) for 5 min, while the other half was kept in the dark. After a further 16 h incubation, 2-(2-methoxy-4-nitrophenyl)-3-(4-nitrophenyl)-5-(2, 4-benzene disulfonic acid) 2h tetrazolium monosodium salt (WST-8) was added into the wells and mixture were incubated for 4 h. After this, 100 μL of dimethyl sulfoxide (DMSO) was added into each well to dissolve the formazan crystal. Eventually, the absorbance was measured at a wavelength of 450 nm using a microplate reader. The cell viability (%) was calculated using the following Equation (1):(1)$$V\%=({\left[A\right]}_{\mathrm{experimental}}-{\left[A\right]}_{\mathrm{black}})/({\left[A\right]}_{\mathrm{control}}-{\left[A\right]}_{\mathrm{black}})\times 100\%$$

The concept of the combination index (CI) was introduced to evaluate the synergistic, additive, and antagonistic effects of two different drugs at different dosages according to the Chou–Talalay method, and the formula for CI is shown in Equation (2):(2)$${\mathrm{CI}}_{50}=\frac{{\left(D\right)}_{\mathrm{DOX}}}{{\left(D_{50}\right)}_{\mathrm{DOX}}}+\frac{{\left(D\right)}_{\mathrm{SQ-}N+L}}{{\left(D_{50}\right)}_{\mathrm{SQ-}N+L}}$$
where (D_50_)_DOX_ and (D_50_)_SQ-N+L_ express the IC_50_ values of DOX alone and SQ-N+L alone, respectively. (D)_DOX_ and (D)_SQ-N+L_ express the concentrations of DOX and SQ-N+L in the combination system at the IC_50_ values. CI > 1 means antagonism, CI = 1 means additive, and CI < 1 means synergism.

### 3.6. Live/Dead Cell Double-Staining Assay

The in vitro anti-tumor efficacy of DOX, POSS-SQ NPs, and POSS-SQ-DOX NPs was evaluated using Calcein-AM/PI (propidium iodide) staining and a CKX 53 fluorescence microscope (Olympus, Tokyo, Japan). First, HeLa cells were incubated with PBS, DOX (12 μg/mL in DMEM, 1 mL), POSS-SQ NPs (18 μg/mL in DMEM, 1 mL), and POSS-SQ-DOX NPs (50 μg/mL in DMEM, 1 mL) in confocal dishes for 4 h. Then, the cells were kept in the dark or irradiated with an 808 nm laser (0.5 W/cm^−2^) for 2 min. And then after 6 h of continuous incubation, the cells were co-stained with calcein AM/propidium iodide to mark viable cells and dead cells, respectively. Cells were imaged with the CKX53 fluorescence microscope via the green/red fluorescence channel.

### 3.7. In Vivo Synergistic Anti-Tumor Effects

Animal experiments were carried out following strict compliance with the provisions of the “Guidelines for the Use and Management of Experimental Animals” (IACUC). In order to evaluate the synergistic anti-tumor effect of chemotherapy and photothermal therapy, we selected mice with a tumor volume of about 120 mm^3^ and randomly divided them into five groups (*n* = 3). The experimental groups were “PBS”, “DOX”, “POSS-SQ +L”, “POSS-SQ-DOX”, and “POSS-SQ-DOX +L”. In total, 200 µL of the PBS, DOX (12 µg/mL), POSS-SQ NPs (18 µg/mL), and POSS-SQ-DOX NPs (50 µg/mL) solution were injected into mice via intratumoral injection and the groups of “POSS-SQ +L”, and “POSS-SQ-DOX +L” were treated with 808 nm laser irradiation for 2 min immediately (tumor site, 0.5 W/cm^2^), and then synchronously imaged every 10 s with a T540 infrared thermal camera (FLIR, Wilsonville, Oregon, USA). After that, all groups were treated with the same operation again every three days, and the tumor volume and body weight of the tumor-bearing mice in each group were measured and recorded for a total of 12 consecutive days. All animal studies were approved by the e Institutional Animal Care and Use Committee guidelines of Shanghai Jiao Tong University.

### 3.8. Histological Studies

All the mice in the three groups were euthanized on the 12th day after therapy. The tumors and main normal organs (heart, liver, spleen, lungs, and kidney) were resected, sliced, and stained. Cell states in the tumor sites and other main normal organs were analyzed using hematoxylin–eosin (H&E) staining; the tissues were immersed in 4% formalin solution, embedded in paraffin after dehydration, and sliced at a 4 μm thickness. For cell proliferation detection, serial tissue sections with a thickness of 4 μm were cut and then stained with an antibody against Ki67 (Servicebio, Wuhan, China) for 1 h. For cell apoptosis detection, the tumor sections were stained with terminal transferase-mediated dUTP nick end labeling (TUNEL) per the manufacturer’s instructions (Promega, Madison, Wisconsin, USA). All slides were digitized with a high-resolution whole-slide scanner (Pannoramic DESK, Kongerizoge Street, Budapest, Hungary).

### 3.9. Blood Routine Assay

Healthy BALB/c nude mice were randomly assigned to two groups, labeled as “PBS” and “POSS-SQ-DOX +L” (*n* = 3 per group). After 12 days, the blood was sampled and detected for the blood routine examination.

### 3.10. Ethical Statement

The animal study protocol was approved by the Ethics Committee for Animal Experiments Affairs in Shanghai Renji Hospital, China (ra-2019-091).

### 3.11. Statistical Analysis

All statistical analyses were performed using GraphPad Prism 9.0 (GraphPad software, San Diego, California, USA). Statistical significance was determined using one-way analysis of variance (ANOVA) for multiple groups. All data are presented as the mean ± SD. *p* < 0.05 was considered statistically significant.

## 4. Conclusions

In summary, POSS-based multifunctional nanoparticles (POSS-SQ-DOX NPs) were rationally designed and successfully applied for chemo-mild photothermal synergistic anti-tumor therapy. POSS-SQ-DOX NPs were strong in stability, good in biocompatibility, and had a high photothermal conversion efficiency, which allowed POSS-SQ-DOX NPs to be used as a PTA and drug transport platform in chemo-mild photothermal therapy. Meanwhile, the calculation of the combination index (0.72, <1.0) indicated the existence of a synergistic anti-tumor effect. Furthermore, due to the better cell permeability and the presence of synergistic therapy, POSS-SQ-DOX NPs could enter into tumor cells more easily, which was conducive to the nanoparticles for more effective mild photothermal therapy. Additionally, chemotherapy also solved the problem of the poor therapeutic effect caused by low temperatures during mild photothermal therapy. Therefore, both in vitro and in vivo experiments showed the effect of chemo-mild photothermal synergistic therapy better than single chemotherapy and photothermal therapy. In conclusion, the developed POSS-SQ-DOX NPs, as a comprehensive nanotherapeutic agent, have great potential to induce chemo-mild photothermal synergistic therapy, improve anti-tumor efficacy and reduce side effects, which is expected to be further applied in clinical tumor treatments.

## Data Availability

Data is contained within the article and Appendix A.

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
