# Peer review of "POSS Engineering of Multifunctional Nanoplatforms for Chemo-Mild Photothermal Synergistic Therapy"

_ijms, 2024, doi:10.3390/ijms25021012_

Round 1
Reviewer 1 Report
Comments and Suggestions for Authors
Gu et al. reported a multifunctional nanocomposite for chemo-mild photothermal synergistic cancer therapy. Overall, this work has been done systematically and supported by data from different dimensions. However, it is undeniable that it is a rough work and there are so many details that need to be explained further. Additionally, there are so many "careless mistakes" that need to be addressed. All the concerns are presented as follows:
1. The authors claim that the aggregation issue affects the photothermal performance of SQ-N and the POSS structure can help to avoid the π-π stacking of SQ-N, thereby maintaining the photothermal effect. However, no experimental results support this point. Additional experiments are needed, such as the evaluation of the photothermal effect of SQ-N aggregates in water and the corresponding heat conversion efficiency.
2. The authors keep saying that PEG could enhance the cell permeability of nanocomposites. It is an “unusual” way to describe the function of the PEG chain. It is more reasonable to say that PEG could enhance the hydrophilicity of nanocomposites and avoid forming random-size aggregates, which is not favorable for cellular uptakes. Please double-check the relevant explanation about the function of PEG regarding cellular uptakes.
3. The introduction section needs to be reorganized, especially in the first paragraph. Why the so-called "MPTT" could synergistically work with chemotherapy?
4. In the main text, some phrases, such as non-toxic, first investigation, significantly superior efficacy, etc., are way too arrogant and not rigorous at all.
5. In Scheme 1, the cell membrane is broken after therapy; however, there is no clue about this according to the results in the manuscript.
6. In the Results and Discussion section, some Figures are "missed", such as Figure S5, Figure 1a...; some Figures are not arranged following the logic of writing. Please double-check the manuscript carefully.
7. Figure S7 demonstrates that POSS-SQ-DOX NPs possess the best hydrophilicity among the other analytes. However, DOX and SQ are hydrophobic molecules (I supposed; that is why the authors want to integrate them into nanoformulations), so it does not make any sense that POSS-SQ-DOX NPs have better hydrophilicity than POSS-PEG. Please explain why.
8. It is very difficult for me to understand a sentence in Page 4 of 15, "To demonstrated (typo) significant chemosensitization effects, it was crucial to maintain tumor site temperature within the range of 39.0-43.0℃, since this facilitates notable enhancement in cell membrane permeability upon increasing temperature from 37.0-43.0℃, thereby promoting nano-drugs uptake by cells." Please rewrite this sentence.
9. In the subsection 2.3., does the authors tried ratios other than 1:2, 1:1, and 2:1? Since 2:1 is the best ratio so far, does it have any chance to achieve a better result with another ratio, such as 3:1 or 4:1?
10. I understand Dox is chemically attached to POSS via the thiol-ene click reaction. It is reasonable to assume that Dox needs to be detached first before exerting its therapeutic functions. Please provide the relevant details about how Dox detached from POSS structure, it is very important.
11. It is absolutely wrong to say that "there was no significant change observed in the body weight of tumor-bearing mice across all groups" on Page 8 of 15 according to the results in Figure 5h.
12. It is way too simple to describe the histological analysis and blood routine results in subsection 2.4.
13. It is suggested to move the general methods part of the supporting information into the Experimental section of the manuscript.
Comments on the Quality of English Language
14. Please double-check the language issues.
Author Response
Thank you very much for taking the time to review this manuscript. Please see the attachment for specific response.

Reviewer 2 Report
Comments and Suggestions for Authors
Reference Report
Title: POSS Engineering of Multifunctional Nanoplatform for Chemo-Mild Photothermal Synergistic Therapy
Manuscript number: ijms-2804854
Submitted to IJMS
By Gu et al
This investigation pioneered the development of a multifunctional nanoplatform for photothermal synergistic therapy. I have identified several areas of concern in this research:
- Introduction: To enhance the introduction of nanomaterials in photothermal therapy, utilizing more recent references like Siddique et al (Nanomaterials 2022;12:2826) and Siddique et al (Nanomaterials 2020;10:1700) would be beneficial.
- Section 2.1: The text of Section 2.1 does not mention Figure 1a.
- Section 2.2: Please rephrase the term "808 nm laser."
- Section 2.2: Clarification is needed regarding why the maximum temperature of 81.8 C significantly exceeds the typical range of photothermal therapy (about 40-43 C).
- Figure 2a and 2b: Error bars should be included for the plots.
- Section 2.3: "Tbale" needs correction to "Table."
- Section 2.3: Please provide a definition for the combination index or include a reference for clarification.
- Table 1: Explanation is required for the absence of data for DOX, SQ-N+L, CI for Free DOX, and Free SQ-N+L.
- Authors might consider discussing the limitations of this study.
- Section 4.5: It would be beneficial to label the Equations.
- Section 4.10: "P" should possibly be revised to "p" according to the manuscript's text.
Comments on the Quality of English Language
No problem to read and understand this manuscript.
Author Response

(The authors gave the same response as above.)

Round 2
Reviewer 1 Report
Comments and Suggestions for Authors
The manuscript can be accepted in its current form.
Comments on the Quality of English LanguageThe language issue has been improved.
Reviewer 2 Report
Comments and Suggestions for Authors
After reviewing the revised manuscript, and the authors responses, I am satisfied with the modifications and corrections as per my comments. The quality of the manuscript is improved.
Comments on the Quality of English LanguageI do not have problem to read and understand the manuscript.